# A Safety Detection Method on Construction Sites under Fewer Samples

**QingE Wu** *[ID], **Wenjing Wang, Hu Chen, Lintao Zhou, Yingbo Lu and Xiaoliang Qian** *[ID]

School of Electrical and Information Engineering, Zhengzhou University of Light Industry, Zhengzhou 450002, China
* Correspondence: wqe969699@163.com (Q.W.); qxl_sunshine@163.com (X.Q.)

**Abstract:** In order to solve the problem of automatically completing safety detection for construction sites and give an alert based on high-speed image streams, this paper proposes a violation of rules and regulations (VoRR) recognition method on a construction site and gives a matching method by automatically obtaining a few samples. The proposed safety detection method consists of five parts, which are redundant information reduction, classification, feature extraction, matching, inference rule and alarm alert. Compared with existing safety detection methods, the accuracy of the proposed method is increased by more than 9%. It not only has better performance, but also has more functions: reminding and alarming. For the subsequent establishment of an unmanned supervision system model on a construction site, this research will provide a new method of decision support, target detection, and recognition in multiple different scenarios.

**Keywords:** violation of rules and regulations; safety detection; redundant reduction; feature matching; inference rule

## 1. Introduction

At present, the main method used for construction site safety is manual on-site supervision or manual viewing of static photos, which makes the manual labor intensity very high. Moreover, due to the limited angle and accuracy of the photos, it is very difficult for the human eye to recognize some subtle problems, meaning that some violation operations cannot be detected in time. Therefore, there is an urgent need for a targeted image intelligence detection system that can reduce manual labor intensity, improve the efficiency of image detection, and effectively extract the detection of violation operations. This paper focuses on the detection of safety glasses, fence crossing, safety helmets, workwear, and safety belts. Properly wearing protective equipment on construction sites can reduce accidents by half. Therefore, the method based on image processing is important for the safety detection of construction sites.

A helmet detection method based on a convolutional neural network for face detection and bounding box regression was proposed by Shen J et al. [1]. Extensive experiments and analyses have shown that the proposed method has considerable advantages in detecting helmet wear. However, the environment was relatively homogeneous, there was no discussion of changing environmental imbalances. Helmet detection methods for YOLOv3, v4, and v5 were studied by Benyang D et al. [2–5]. The experimental results showed that the detection speed was improved, which met the real- time requirements of the helmet detection task, but the detection accuracy still needs to be improved. A quadratic template matching algorithm for the fast recognition of target images was proposed by Wu G et al. [6]. By applying the algorithm to the recognition of electric power equipment and the detection of abnormal states, it was found that the matching algorithm can not only accurately locate and identify electrical equipment, but also detect equipment faults. Compared with other commonly used template matching algorithms, the matching speed is much faster, but the recognition of small targets was very poor. A weakly supervised real-time target

detection method was proposed by Hongkai Yu et al. [7]. This method used image-level annotation, which did not need to go through the acquisition process of the target candidate sets. Based on the high-precision and high-speed center net object detection algorithm, a helmet-wearing detection method incorporating novel features was proposed by Huang Li et al. [8]. The experimental results show that the method was based on an improved loss function combined with pixel feature statistics for image processing. Helmet wearing can be accurately determined, and the overall average detection accuracy is verified, but there was no detection of anomalies for other states. A convolutional neural network for helmet recognition based on bidirectional features was established by Tianyu Li et al. [9]. The comparison of experimental results showed that the recognition rate of fuzzy and small helmets was improved. A helmet-wearing detection method based on head region localization was proposed by Yuwan Gu et al. [10]. The method is based on an open pose estimation model. Information about the construction personnel is obtained by introducing residual network optimization features, and the head region is determined by the three-point localization method. Helmet wearing is determined through correlation detection of the head region. However, no solution to the problem of environmental imbalance is given for helmet-wearing detection.

The effectiveness of existing perceptual hashing algorithms in detecting content change image operations was studied by Samanta P et al. [11], and DCT-based hashing algorithms achieved good results in classifying content retention modifications and content change operations, but there was no detection of anomalies. A local binary pattern histogram face recognition system based on UAV technology was proposed by Wang L et al. [12]. This system can detect the ideal person by using a pre-trained LBPH face recognizer to identify the person in the acquired frame. The image edges are affected by halo artifacts nearby in some special environments. Image noise cancellation, color wavelength compensation, and other processing methods were utilized by Zhang H et al. [13–16] to eliminate speckle noise and enhance image details. This not only preserved the edges and effectively reduced the halo artifacts, but also had a better effect on image edge smoothing. The "Summit Navigator" method proposed by Dinh T H et al. [17], and the histogram-based hyper-spectral image segmentation algorithm proposed by Chakraborty R et al. [18], which can effectively extract the local maxima of the image histogram, but did not work well for differentiating objects of different sizes and states. A lightweight deep learning architecture Cloud Seg Net was proposed by Dev S et al. [19], which was the first image segmentation framework for daytime and nighttime images containing clouds, but it required a large number of samples for learning and the detection speed is slow. Conventional bilinear convolutional neural networks are subject to overfitting problems due to their many parameters and high complexity. A virtual artificial intelligence environment was modeled by Lee Jaekyu et al. [20–22]. This paper presents the overall application development methodology. It includes the structure and methods for collecting construction site image data, the structure of the training image dataset, the methods for expanding the image dataset, and the artificial intelligence backbone model applied to migration learning, but the detection speed needs to be improved. Chung William Wong Shiu et al. [23,24] designed a monitoring system-based innovative safety model to provide real-time monitoring of construction site personnel and environment, and the proposed model identified real-time personnel safety problems. Yahu, Y. et al. [25–29] proposed a video image anomaly detection method, but there is no alarm alert for classification of anomalies. The detection method proposed by Kamoona, A.M. et al. [30–36] for the identification of fewer species. However, how the redundant information reduction and quantification is performed affects the performance in dimensionality reduction techniques, which is necessary for detecting construction site conditions through image noise reduction, but none of the existing literature addresses this, which is the key to this paper's research.

Artificial neural networks are abstracted from the neuronal networks of the human brain in terms of information processing. By building some simple models, different networks with different connections are formed. An operational model is formed by in-

terconnecting a large number of neuron nodes. Each node represents a specific output function, called the activation function. Each connection between two nodes represents the weighted value of the signal passing through that connection, called the weight, which is equivalent to the memory of an artificial neural network. The output of the network varies depending on the connections, the weight values, and the activation function. Convolutional neural networks are a class of feed-forward neural networks with deep structure. Both supervised and unsupervised learning can be performed. The shared parameters of the convolutional kernel within the hidden layers and the sparsity of the connections between the layers enables convolutional neural networks to lattice-point the features with less computational effort. Convolution is the simple process of applying filters to the input content. It results in activation expressed in numerical form. By applying the same filter repeatedly to the image, an activation map called a feature map is generated. This represents the location and intensity of the detected features. In this paper, based on the neural network, a safety detection method for the violation of rules and regulations (VoRR) on construction sites is proposed.

## 2. Safety Detection Method

The procedure of the safety detection method proposed in this paper consists of seven steps, and then the network of the procedure consists of seven layers, as shown in Figure 1. The detailed procedure is as follows: the input layer is the different video images of the construction site, and the detailed discussion of the processing are presented in Section 3 to Section 8. The second layer is to eliminate the background and non-target objects in the image of the site. The third layer is to classify the obtained necessary target by a previous layer, i.e., the classification of normal, abnormal, and various violations, then compile the codes for different violation categories labeled as feature indexes. The fourth layer is to calculate the feature index to obtain the feature value of the violation type by a previous layer. Based on the feature value calculated by a previous layer, the fifth layer completes the feature matching between the features to be detected and the known features stored in this layer. The sixth layer gives the inference rules and carries out the decision inference based on the matching results obtained by a previous layer and national industry standards. The last layer is the output layer, the result of which is the violation type. From the captured image input, the violation case on the construction site is obtained by this method. This method solves the problem in that the violation type can be found automatically through the on-site video, but it does not need manual supervision to carry out safety detection, which achieves unmanned management. The method proposed in this paper not only reduces the labor intensity and the subjectivity of human judgment, but also solves the contradiction between supervisors and workers.

As shown in Figure 1, this method consists of five main functional modules to form an intelligent safety detection recognition, i.e., redundant information reduction, classification of VoRR, feature index calculation, feature matching of VoRR, and inference rules. When the final detection result is obtained, the functions of security detection and alarm reminder are realized. Through the large data stream acquisition design of the image, reduction, feature extraction, automatic matching, and inference rule detection are achieved under fewer samples. This paper realizes the correlation analysis of historical data, based on the combination of each node of the massive data. Big data analysis was completed to obtain the most suitable adjustment scheme for the automatic security detection system. By helping the staff to carry out the most appropriate monitoring and management, the functions of alarm and reminder can be realized. In addition, the methods in this paper make different levels of security detection more suitable for different objects, and different environments. It can be used safely, securely, and conveniently for video detection in security departments.

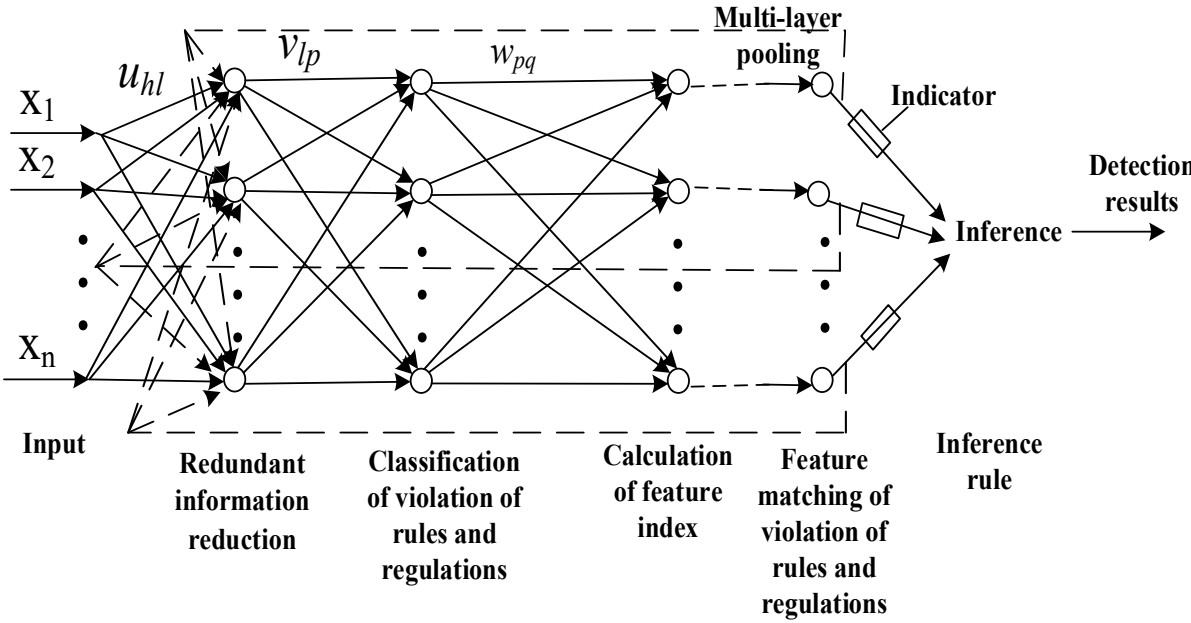

**Figure 1.** Framework model for safety detection.

Figure 1 shows the different intelligent detection modules that will be discussed the next sections. Section 3 proposes a reduction algorithm for redundant information based on the lightness of the attributes. It is to save detection time and improve detection accuracy. Section 4 proposes a classification method of VoRR for the diversity of construction site violations. The initial large target classification is established based on the feature attribute vector to be detected. Section 5 proposes a feature extraction method. The singular value matrix is used to calculate the feature vector of the object to be detected in the image. Section 6 proposes a feature matching algorithm. The features of the target to be detected are implemented to match the standard library by different similarity matching so as to match the target parameter vector. Section 7 establishes inference rules as the criteria for identifying targets and determining violations.

## 3. Redundant Information Reduction

### 3.1. Reduction Algorithm

To determine whether an element or a target set is a necessary element or a necessary target area of the construction site, the following Formulas (1) and (2) can be used to calculate the magnitude of the correlation in order to make a judgment or reduce the redundant information based on the size of the attributes. This method is called the correlation degree method. From the intelligent detection framework model shown in Figure 1, in the redundant information reduction layer, there are $M$ filters that carry out the semantic association calculation and calculate the association degree using the association operator, while adjusting the weights $u_{hl}$ of the input layer, thus providing better processing of the image, where $0 \leq u_{hl} \leq 1$. The data derived from this layer filter show that the adjustment of $u_{hl}$ is as follows: If the image in the window has more feature information, the value of $u_{hl}$ is increased; otherwise, the value of $u_{hl}$ is decreased. Here, $h = 1, 2, \cdots, N$ is the number of inputs and $l = 1, 2, \cdots, M$ is the number of reduction filters. The input value $S_l^t$ of the reduction layer at a time $t$ is $S_l^t = b_l + \sum_h u_{hl} I_h^t$, where $b_i$ is an adjustable constant and $I_h^t$ is the input image information value.

Let $m_1(X_L)$ and $m_2(X_U)$ be the believability of the lower approximation $X_L$ and the upper approximation $X_U$ of the violation event $X$, respectively, while the calculation of believability and $m_2(X_U)$ is the calculation of the probability distribution function, which can also be assigned by experiments or experts to a violation by the magnitude of the objective phenomenon, but requires that the sum of the believability for all events

$X$ is 1. Additionally, the total believability $m(X)$ can be calculated by $m_1(X_L)$ and $m_2(X_U)$, which is

$$\min\{m_1(X_L), m_2(X_U)\} \leq m(X) \leq \max\{m_1(X_L), m_2(X_U)\} \tag{1}$$

The final specific value is generally given by the expert according to Formula (1). Alternatively, in the case of multiple experts, e.g., $n$ experts assign values to the same violation event $X$, then the basic trust degree $m(X)$ is calculated using the following formula:

$$m(X) = \sum_{i=1}^{n} \omega_i m_i(X) \tag{2}$$

Here, $m_i(X)$ is the believability of the $i$th expert in the assignment of event $X$, the weight value of the $i$th expert, and $0 \leq \omega_i \leq 1$. According to people's natural habit of dealing with uncertain information, $\omega_i$ is calculated by an axisymmetric function.

Based on the calculation of the basic believability above, the range of true values is further narrowed, and the judgment result is finally obtained. The method is that the trust function for $X$ is $m^*(X)$. If, in the set $X$, the set after removing an element is set to $Y_1$, trusted to $m^*(Y_1)$ and $|m^*(X) - m^*(Y_1)| < \varepsilon$, then the element is considered to be removable, where $\varepsilon$ is a predetermined threshold value. Repeat this process until a certain subset $Y_K$ has no elements that can be removed, then $Y_K$ is the final filtering result, and the reduction result is input to the next module layer.

The defining formula is as follows:

$$S = -\sum_{i=1}^{n} P(\omega_i|x) \log P(\omega_i|x) \tag{3}$$

Here, $n$ is the violation category number, $x$ is the construction image feature, and $\omega_i$ represents the $i$th category of violation. For an image of size $M \times N$, the information loss is defined as follows:

$$S = -\sum_{k=0}^{G-1} P_k \log P_k \tag{4}$$

Define $P_k$ as follows:

$$P_k = \frac{1}{MN} \sum_{i=0}^{M-1} \sum_{j=0}^{N-1} \rho_{ij}(k) \tag{5}$$

$$\rho_{ij}(k) = \begin{cases} 1, & R(i,j) = k \\ 0, & \text{other} \end{cases}, k = 0, 1, \cdots, G-1 \tag{6}$$

where $G$ is the number of gray levels in the violation screen, $R(i,j)$ is the gray value, and $P_k$ satisfies Formula (7):

$$\sum_{k=0}^{G-1} P_k = 1 \tag{7}$$

The relative information loss can measure the degree of information loss. If the information loss of the sample image is $S_1$, and the information of the sample image at the sampling rate of $\eta$ is $S_\eta$, the relative information loss is shown in Formula (8):

$$\sigma_{1-\eta} = \left| \frac{S_1 - S_\eta}{S_1} \right| \tag{8}$$

From the above analysis, it can be seen that the sampling rate can be chosen according to the relative information loss.

### 3.2. Simulation of Redundant Information Reduction

Because the construction site scene is complex, there will be some interference objects. In order to verify the practical performance of the proposed redundancy information

reduction algorithm, simulation experiments of the proposed algorithm are carried out. By reducing the redundant information, information not related to the construction site can be filtered out of the images, and only the construction personnel information can be retained. The filtering effect is shown in Figure 2.

(a) original image              (b) remove irrelevant information region

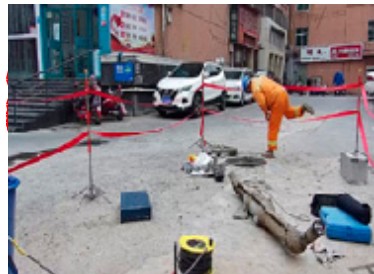 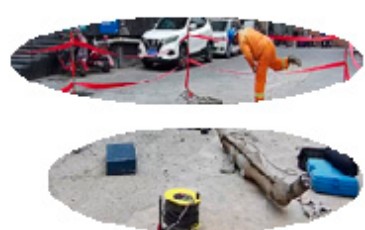

**Figure 2.** Filtering effect for irrelevant information on construction site.

As shown in Figure 2b, which is a captured image of a construction site, after the redundant information reduction, the image background and some image target components unrelated to construction are filtered out, while the construction personnel information and construction personnel operation tools are retained. This shows that the given correlation degree approximate reduction algorithm is feasible and reliable.

Furthermore, for the noisy image, noise reduction can also be achieved using the Sobel operator and the Prewitt operator.

The Sobel operator is a first-order differential operator. It calculates the gradient of each pixel using the gradient values of the neighboring regions of the pixel. The detection of edges is based on the phenomenon of the grayscale between the upper, lower, left, and right neighbors of an image pixel point, while the weighted difference reaches its extreme value at the edge. It is given by the following formula:

$$S = \left(dx^2 + dy^2\right)^{1/2} \tag{9}$$

The Sobel operator is an $3 \times 3$ operator template. Figure 3 shows the 2 convolution kernels $dx, dy$. The maximum value of the two convolutions is the output value at that point. The result is an edge magnitude image. The edges are picked up according to a certain threshold value. The algorithm has a smoothing effect on noise and can provide more accurate edge orientation information, but the accuracy of edge localization is not high enough.

| $-1$ | 0 | 1 |
|------|---|---|
| $-2$ | 0 | 2 |
| $-1$ | 0 | 1 |

| 1 | 2 | 1 |
|---|---|---|
| 0 | 0 | 0 |
| $-1$ | $-2$ | $-1$ |

**Figure 3.** Sobel operator.

The idea of the Prewitt operator edge detection is similar to that of the Sobel operator. The Prewitt is given by the following formula:

$$S_p = \left(dx^2 + dy^2\right)^{1/2} \tag{10}$$

The two convolution kernels dx, dy shown in Figure 4 form the Prewitt operator. Its differential operations are defined in an odd-sized template.

| | | | | | |
|---|---|---|---|---|---|
| − 1 | 0 | 1 | 1 | 1 | 1 |
| − 1 | 0 | 1 | 0 | 0 | 0 |
| − 1 | 0 | 1 | − 1 | − 1 | − 1 |

**Figure 4.** Prewitt operator.

The algorithm template is convolved with the image pixel gray values from left to right and from top to bottom in order. The operator has a smoothing effect on noise, but the localization accuracy is also not high enough.

According to the above redundant information reduction method, simulation is performed and then compared with other noise reduction algorithms. The results can be seen in Figure 5.

(a) image with noise
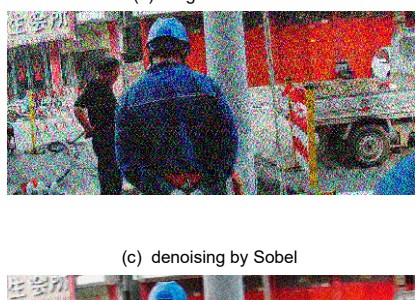

(b) denoising by Prewitt
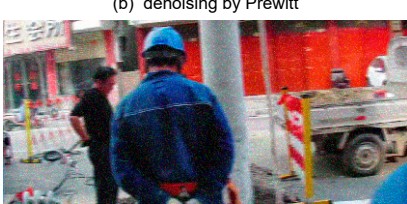

(c) denoising by Sobel
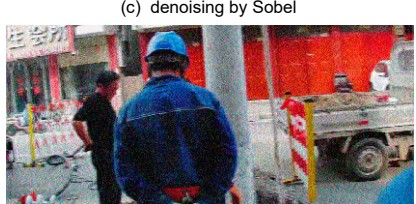

(d) denoising by proposed reduction
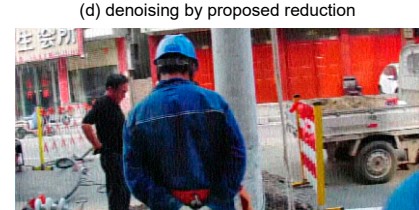

**Figure 5.** Filtering effect for noise.

From Figure 5, Figure 5a is the image with added noise, and Figure 5d is processed using the noise reduction method proposed in this paper. Compared with Figure 5b,c, it can be seen that the noise reduction method in this paper is superior to other methods in terms of noise reduction, and almost all the noise is filtered out. It shows that this redundant information reduction method is feasible and effective.

## 4. Establishing a Similarity Classification Method for VoRR

### 4.1. Classification Method

In the VoRR classification module layer, there are $N$ filters and weights $0 \leq v_{lp} \leq 1$ from the previous reduction layer to the current classification module layer. The adjustment of $v_{lp}$ is as follows: the data from this filter layer show that if the variance of the feature is less than a given threshold, the value of $v_{lp}$ is increased; otherwise the value of $v_{lp}$ is decreased. Here, $l = 1, 2, \cdots, M$ and $p = 1, 2, \cdots, N$ are the number of filters in the reduction and classification layers, respectively. The input value $C_p^t$ of the classification layer at a time $t$ is $C_p^t = \sum_l v_{lp} P_l^t$, where $P_l^t$ is the output data value of the reduction layer.

Firstly, the construction site scene is complex and construction tools are in every place. Establishing an initial large target classification can identify violations more precisely; large targets on construction sites are classified according to the requirements of target features or rules. The helmets, belts, work clothes, warning vests, protective glasses, and illegal operations are first classified using image feature extraction methods. Examples include straddling security bars and fences, crooked hats, wrong belts, and incorrect welding methods. Then, a vector of standard safety marker features is created, which is composed of gray mean, variance, feature shape, perimeter enclosed by the shape, area, etc.

For the collected construction site images from different places, the images are classified according to source, format, performance, scene, and usage. Then, indicator features are given, and the data classification is achieved by detail classification of features of the same class. The classification of detail features is based on the degree of proximity between feature attribute values, i.e., the degree of membership. If the differences between two attribute values are less than a given threshold, it is decided that the two feature attributes belong to the same class; otherwise, they belong to different classes.

The so-called data classification is to classify the feature attribute vector to be detected into the most similar category composed of known attributes. The classification method given below is referred to as the nearest neighbor method.

Consider $X = (x_1, x_2, \cdots, x_m)$ of the target to be detected as a point in the $R^m$ space. In order to know their association from the number of features, the nearest $c$ known categories $X_l$ to the target $X$ to be detected can be obtained, which is denoted as $N_c(X)$. Where $X_l$ denotes the $l$th category, which has the same number of features as $X$ in the known category data $l = 1, 2, \cdots, c$. Assume that the $m$ feature attributes of the target to be tested are $(x_1, y_1), (x_2, y_2), \ldots, (x_m, y_m)$, where $x_i$ denotes the feature and $y_i$ is the category corresponding to the feature. The values of the category are 0 and 1, i.e., if the feature detected matches the features of a known category, it is labeled as 1; otherwise, it is labeled as 0. For a given target $X$ and $l$th category $X_l \in N_c(X)$ to be examined, the category of $X$ can be determined using Formula (11):

$$y = \frac{1}{m} \sum_{j=1}^{m} y_j \tag{11}$$

where, if the value $y$ is greater than the given threshold $\varepsilon$, $X$ belongs to the $l$th category and is labeled as 1; when it is less than $\varepsilon$, $X$ does not belong to the $l$th category and is labeled as 0.

To improve the accuracy of the nearest neighbor method, a weighted value can be given in Formula (11), which is called the weighted nearest neighbor method. In this algorithm, the calculation of decision weights is introduced for each instance. For the instance $X$ to be examined, the category $X_l = (x_{l1}, x_{l2}, \cdots, x_{lm})$ is known. The distance $d_1, d_2, \cdots, d_c$ between the $X$ and $c$ nearest neighbor vectors is defined as follows:

$$d_l = \|X - X_l\| = \sqrt{\sum_{j=1}^{m} \left( x_j - x_{lj} \right)^2}, \ l = 1, 2, \cdots, c \tag{12}$$

Since there are $c$ nearest neighbor vectors in the nearest neighbor domain $N_c(X)$ of $X$, according to the meaning of the distance and weight between the $c$ vectors, and based on the weight $\omega_l$ to satisfy $\sum_{l=1}^{c} \omega_l = 1$, the decision weight $\omega_l$ can be defined as:

$$\omega_l = \frac{1}{c-1} \left( 1 - \frac{d_l}{\sum_{l=1}^{c} d_l} \right) \tag{13}$$

Based on the weighted nearest neighbor decision rule, Formula (11) can be modified as:

$$y = \frac{1}{m} \sum_{j=1}^{m} \omega_j y_j \tag{14}$$

If the value $y$ is greater than the given threshold $\varepsilon$, then $X$ belongs to the class $l$. Otherwise, $X$ does not belong to the class $l$. Then, it is input into the next function module.

After determining the $c$ nearest neighbors of the instance $X$, the decision weights of the components in the instance are calculated based on the magnitude of the decision weights, judging the magnitude of the role of its components in predicting the class affiliation of $X$.

The algorithm proposed in this paper is a very effective method. It has high noise resistant in the training data, and is also very effective when given a large enough training set. The influence of isolated noise samples can be eliminated by the weighted average of the $c$ nearest neighbors.

### 4.2. Simulation of Classification Methods

To verify the practical performance of the proposed target classification algorithm, a simulation comparison with existing classification algorithms [9,12] for image information classification is presented here. Additionally, among the current classification methods, these two methods are open and unrestricted. Other methods are using national standards plus human eye recognition, which is not the better method at present. The simulation results are shown in Figure 6.

(a)  Classified target by proposed method    (b) Extract selected area by this method

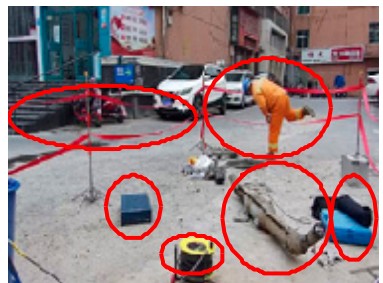
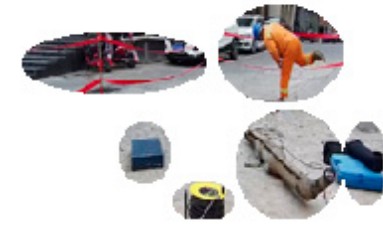

(c) Hashing algorithm                        (d) MobileNet-SSD

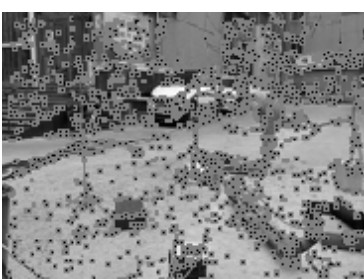
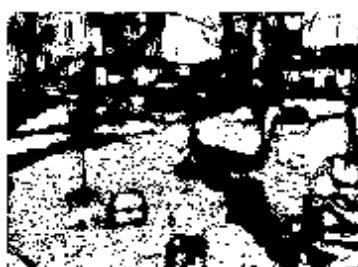

**Figure 6.** Effectiveness of the classification algorithm proposed for the classification of construction site targets.

As can be seen from Figure 6, the classification algorithm proposed in this paper can classify the target regions well, as shown in Figure 6a,b. The algorithm can classify images composed of different objects with high accuracy. Although the segmented targets have a little redundant information, the elements are identified with less noise, and almost all the regions with the features of the targets of interest are classified. At the same time, the selected targets were extracted with little information loss and well-detailed features were maintained. However, as can be seen from Figure 6c,d, the feature points

of these two methods did not detect the target objects for safe construction. The hashing algorithm detected the target information, but also mixed with some irrelevant information. The MobileNet-SSD did not detect the safe target objects, and some target objects were incompletely detected. Both methods lose the color information. The method used is that the defined function carries out convolution with all the feature information in the image, so that the target features have convolved values that are more obvious. The results of classification from other algorithms showed that more original information is lost, almost all color features are missing, and the boundaries of classification are not obvious. There is a lot of noise, and some small targets with distinctive features are overwhelmed by noise.

*4.3. Comparison between the Proposed Methods and the Existing Classification Methods*

In addition, to evaluate the accuracy of the proposed classification method, it can be defined as follows.

Assuming the total number of experiment statistics is $T$, the number $n$ of correct classified statistics is counted by the counting program, then the correct classified rate $R$ is defined as:

$$R = \frac{n}{T} \tag{15}$$

For 300 statistical experiments, repeat 80 times, and use the counting program to count the effective responses $n$ at each time. Then, the correct classified rate is calculated using the Formula (15), as shown in Figure 7.

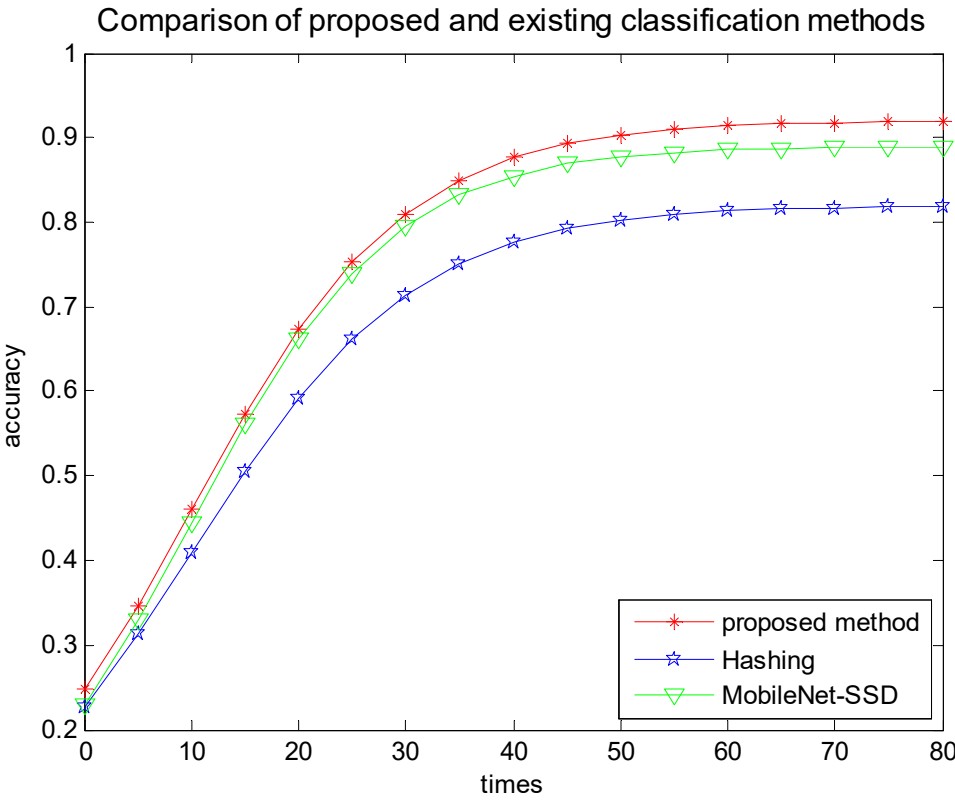

**Figure 7.** Comparison of proposed and existing classification methods.

Comparing the accuracy of the proposed method with the existing hashing algorithm and MobileNet-SSD, the experimental results show that the accuracy of the proposed method is 91.82%, that of the hashing algorithm is 80.78%, and that of MobileNet-SSD is 88.26%, as shown in Figure 5 and Table 1. In addition, the classified speed of the proposed method is 1.18 ms. However, that of hashing is 3.41 ms and MobileNet-SSD is 2.56 ms.

**Table 1.** Comparison of proposed and existing classification methods.

| Classification Methods | Accuracy (%) | Classified Speed (ms) | Anti-Interference Ability |
|---|---|---|---|
| Proposed method | 91.82 | 1.18 | strong |
| Hashing | 80.78 | 3.41 | weak |
| MobileNet-SSD | 88.26 | 2.56 | weak |

Although the hashing algorithm and MobileNet-SSD have been used for target classification, from Figure 7 and Table 1, the proposed correct classified rate is higher than that of the existing methods, and the running time is smaller. Knn classification is just one of the main methods used in this paper for classification between different classes. However, in the actual implementation of the classification process, other auxiliary means such as small probability events, the importance of the appearance of violation targets in the field, the differentiation of violation cases calculated by the adversarial network differentiation function, and Yolo v5 tracking are also needed to finally achieve the classification of violation targets. The single Knn method cannot classify multiple violation cases.

## 5. Proposed Inner Product Singular Value Decomposition Method for Feature Extraction

Image features reflect discontinuities in the local features of an image, which marks the end of one region and the beginning of another. Using the method of affiliation function, the anomalous affiliation of the feature points to be detected can be calculated and the effect of discontinuities in the local features of the image can be easily detected. The values of the parameter indexes from the extracted features are obtained by the decomposition of the singular matrix. Here, the affiliation function and the singular matrix are given. The method is as follows.

In the module layer of feature metric calculation, $Q$ filters are different calculators, and the weights from the classification layer to the feature metric calculation layer are $0 \leq w_{pq} \leq 1$. The adjustment of $w_{pq}$ is as follows: If the similarity of the corresponding metric values is greater than the given threshold, the value of $w_{pq}$ is increased; otherwise, the value of $w_{pq}$ is decreased, where $p = 1, 2, \cdots, N$ and $q = 1, 2, \cdots, Q$ are the number of filters in the classification and metric calculation layers, respectively. The input value $F_q^t$ of the indicator calculation layer at a time $t$ is $F_q^t = \sum_p w_{pq} E_p^t$, where $E_p^t$ is the output data value of the classification layer.

Feature extraction is to extract the features of the target information from the image, and calculates the feature vector of the target to be detected. According to the characteristics of the detection target itself and expert experience, some feature parameters of the violation are first extracted. Then, these features are mined separately for their feature points $A$. Examples of feature points are whether workers wear helmets and safety belts, and normality, abnormal behavior, and abnormal operating parts during construction. Given the affiliation function as follows in Formula (16), the affiliation degree $\mu_A$ of the feature point anomaly $A$ to be detected is calculated by this function. The feature point, called core point $A_0$, is given by historical data or domain experts, and its affiliation degree is noted as $\mu_{A_0}$. By defining the affiliation degree between $A$ and $A_0$, the affiliation degree determines whether the point $A$ to be detected is a feature point. In this way, the key feature parameter vector $\Lambda = [\xi_1, \xi_2, \cdots, \zeta_n]^T$ of the anomaly is obtained, where $\xi_i = [\xi_{i1}, \xi_{i2}, \cdots, \xi_{ik}]^T, i = 1, \cdots, n$. $n$ denotes that there are $n$ behavior classes. Each behavior class has $k$ parameters for the key features and $i$ denotes the $i$th category of behavior.

To calculating the affiliation of the anomaly $A$, assume that $\xi_{ij}$ is the $j$th parameter indicator value of the class $i$ violation feature, which is extracted by the singular value decomposition of Formula (17) below. Then, its degree of affiliation $\mu_{ij}(A)$ corresponding

to $A$ can be calculated by a continuously derivable function with good performance, which is defined as follows:

$$\mu_{ij}(A) = \beta \frac{1 - e^{-\alpha \xi_{ij}}}{1 + e^{-\alpha \xi_{ij}}} \tag{16}$$

where $\alpha, \beta > 0$ is a constant that controls the slope of the affiliation curve.

To extract the violation features by a singular matrix, a special matrix needs to be constructed, and a sequence of violations of length $k$ is defined as $y = \{x_1, x_2, \cdots, x_k\}$. A matrix $D$ of $m \times n$ can be constructed from $y$, and the construction $D$ is as follows:

$$D = \begin{pmatrix} x_{11} & x_{12} & \cdots & x_{1n} \\ x_{21} & x_{22} & \cdots & x_{2n} \\ \vdots & \vdots & & \vdots \\ x_{m1} & x_{m2} & \cdots & x_{mn} \end{pmatrix} \tag{17}$$

After obtaining the processed data, the singular value decomposition is performed. The signal sequence is $y = \{x_1, \ldots, x_{25}\}$, and $x_i(i = 1, \ldots, 25)$ is the $i$th point. Define the construction matrix A ($21 \times 5$) as follows:

$$A = \begin{pmatrix} x_1 & \cdots & x_5 \\ \vdots & \ddots & \vdots \\ x_{21} & \cdots & x_{25} \end{pmatrix} \tag{18}$$

Because of the differences between different features, the singular values obtained are also different. Choosing the first 25 dimensions as the feature vector, $x_1$ refers to the normal helmet-wearing quantified feature value 1, and $x_2$ refers to the feature value 0.9. According to the national requirements of the helmet-wearing standard, this paper quantified the value of 1, 0.9 ......, etc. There are 25 features index values extracted.

The constructed regular matrix associated with the violation must satisfy the following condition: there are two orthogonal matrices $U, V$, the formulas of which are as follows:

$$\begin{aligned} U &= [u_1, u_2, \ldots, u_M] \in R^{(M \times M)}, \\ V &= [u_1, u_2, \ldots, u_N] \in R^{(N \times N)}, \\ D &= [\text{diag}[\lambda_1, \ldots, \lambda_Q], 0](Q = \min\{M, N\}) \end{aligned} \tag{19}$$

where the U matrix is $m \times m$, the V matrix is $n \times n$, and $\lambda_1, \lambda_2 \cdots \lambda_Q$ are the singular values obtained by decomposing the matrix $D$. Referring to the edge points and inflection points on the violation image, their relationship is $\lambda_1 \geq \lambda_2 \geq \cdots \geq \lambda_Q$.

The singular values obtained by decomposing the singular values that can be used as the eigenvalues of the violation features, so that after acting on the matrix $D$, a diagonal matrix is obtained as follows:

$$\begin{pmatrix} \lambda_1 & 0 & \cdots & 0 \\ 0 & \lambda_2 & \cdots & 0 \\ \vdots & \vdots & \lambda_r & \vdots \\ 0 & 0 & \cdots & 0 \end{pmatrix}_{m \times n} = U \begin{pmatrix} x_{11} & x_{12} & \cdots & x_{1n} \\ x_{21} & x_{22} & \cdots & x_{2n} \\ \vdots & \vdots & & \vdots \\ x_{m1} & x_{m2} & \cdots & x_{mn} \end{pmatrix} V^T \tag{20}$$

where $r = \min\{m, n\}$ and $\lambda_1, \lambda_2, \cdots, \lambda_r$ is the singular value obtained by decomposing the matrix $D$. For each violation, for example, helmet, safety belt, goggles, protective clothing, isolation belt, operation violations, etc., set five indicator values. Taking helmet wearing as an example, these are positive wearing, slanting wearing, crooked wearing, reverse wearing, and with or without buckle cap belt, corresponding to $x_{ij}$ as the characteristic indicator value of each violation.

This diagonal matrix Formula (17) is used to perform the inner product operation on the class *i* violation target $I_i$, that is

$$[\xi_{i1}, \xi_{i2}, \cdots, \xi_{ik}] = [\text{diag}[\lambda_1, \lambda_2, \cdots, \lambda_r], 0] \otimes I_i \tag{21}$$

Here, after decomposing the class *i* violation target with the singular value matrix, $\xi_{i1}, \xi_{i2}, \cdots, \xi_{ik}$ are the value of *k* parameter indicators obtained, which is the extracted target feature. It is finally input to the next module layer.

To verify the performance of the proposed inner product singular value decomposition algorithm for extracting features, several different image feature extraction algorithms [7,16] were selected to simulate the feature extraction of the helmet. The simulation results are shown in Figure 8.

(a) original image

(b) extraction by proposed algorithm

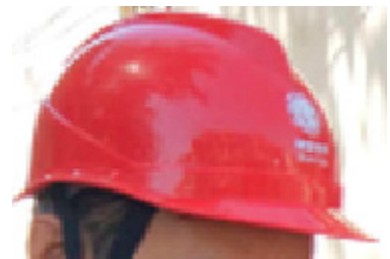
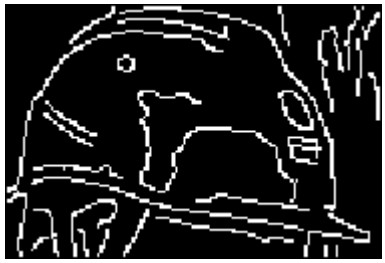

(c) extraction by steering kernel

(d) extraction by weak supervision

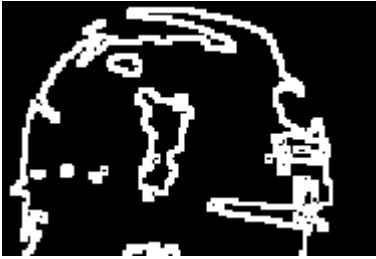
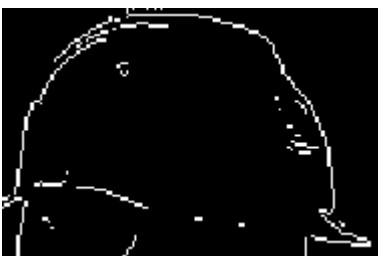

**Figure 8.** Comparison of the effectiveness of the proposed and existing feature extraction algorithms for feature extraction of a helmet.

As can be seen from Figure 8, compared with other algorithms, the feature extraction algorithm proposed in this paper has the best effect on feature extraction, which has clear edge contours and less noise. Almost all features are extracted to maintain good detailed functionality, as shown in Figure 8b. In Figure 8c, there is too much extracted redundant information, and a small amount of feature information is lost. In Figure 8d, there is less redundant information extracted, but too much useful feature information is lost.

## 6. Proposed Association Matrix Method for VoRR Feature Matching

### 6.1. Feature Matching Method

In the feature matching module layer, *Q* filters are pooling operators. Different multiplicities are applied to the image, then the features and their metrics are iteratively computed at different multiplicities to further compute the fine features. By using the pooling operator, the similarity $r_{ij}$ between the minutiae features and metric values is obtained, the *m*th fetch of the known class *i* features in the direction of the *j* index parameters are calculated. Then input to the next module layer.

Feature matching is performing difference matching or similarity matching between the features of the standard library and the target features to be examined. The output value $R_i = (r_{ij})$ of the feature calculation layer at the moment of $t$ is used to correlate and classify diverse features, different indicators, and true/false features, which is to match the target parameter vector. The matching method for the correlation matrix is given as follows:

For the established set of associated data labels, the $m \times n$ association matrix $R$ is established, where $m$ is the number of target sets in the current frame and $n$ is the number of target sets in the previous frame. The calculation of the association matrix $R(i, j)$ is defined as follows:

$$R(i,j) = \begin{cases} |r_i - r_j|, & \text{if } r_i + r_j > \sqrt{(x_i - x_j)^2 + (y_i - y_j)^2} \\ \infty, & \text{else} \end{cases} \tag{22}$$

where $r_i$ is the size of the $i$th dataset, $r_j$ is the size of the $j$th dataset, $(x_i, y_i)$ is the center of the $i$th dataset, $(x_j, y_j)$ is the center of the $j$th dataset, and $\infty$ is a large value. $R(i, j)$ is the similarity comparison by comparing the recognized target image feature vector with the target vector in the national standard library. The vectors in the national standard library are known categories that have been trained and stored in the retrieval system.

The matching matrix is used to match the current data set with the previous data set. First, the element with the smallest value instead of $\infty$ is selected in the matching matrix $R$. The rows and columns corresponding to this element are the numbers of the current dataset and the previous dataset, respectively, so that the dataset corresponding to the rows matches the dataset corresponding to the columns. Then, all the element values of the completed matched rows and columns are changed to $\infty$. Continue to search for the minimum value in the matching matrix $R$ to complete matching the dataset until all values in the matrix become $\infty$. At the end of the search, rows with no matching data are found to represent the appearance of a new dataset in the current dataset, and columns with no matching data found represent the disappearance of a dataset in the current dataset.

The size of $R(i, j)$ is calculated using Formula (22) above, given a threshold value $\mathbb{R}$, if the following formula is satisfied:

$$R(i, j) < \mathbb{R} \tag{23}$$

Then, the $i$th target set and the $j$th target set are closely related, and the two target sets are judged to match; otherwise, the two target sets do not match. By this method, as long as all the target sets are judged, the matched data and the unmatched data will eventually get. Then input this result to the next module processing layer, that is, the inference layer.

### 6.2. Experiment and Analysis

There are two images; one highlights the target and the other highlights the background, which is not related to the construction. Now, the target and background areas are matched separately using the feature matching method given in this section. If the match is successful with high probability, the algorithm automatically implements the fusion of the two images and matches the fusion result with the original image again, and the successful match image is recorded. Here, 50 images were randomly screened out for two matches for a total of 1225 times, resulting in 1117 correct matches, with a correct rate of 91.18%. The results are shown in Figure 9.

It can be seen from the comparison between Figures 9c and 9d that the matching of Figure 9a,b is successful. After the fusion of Figure 9a,b, the information in Figure 9c is the same as that of the original image in Figure 9d. The matching accuracy is higher than 90%, which indicates that the matching method proposed in this paper is feasible and effective.

(a) target area

(b) background area

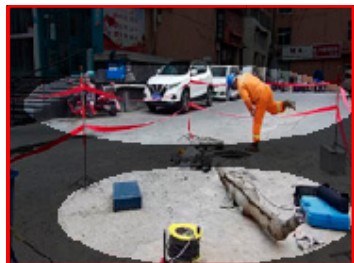

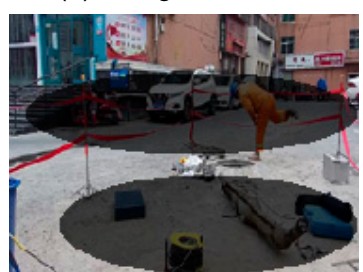

(c) matching fusion

(d) original image

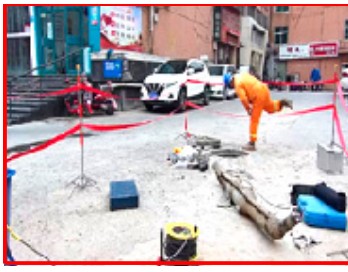

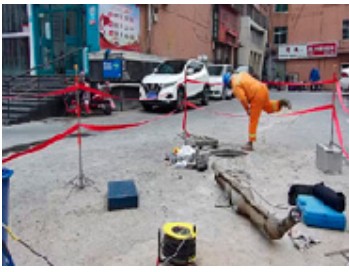

**Figure 9.** Matching results of the proposed correlation matrix matching method for two images.

## 7. Inference Rule Establishment

### 7.1. Inference Rule

The filter of the inference rule module layer is the criterion for identifying the target and judging the violation. The output value $S_i = (s_{ij})$ of the matching layer at a time $t$ is used to establish the violation judgment rule to realize the functions of safety detection, warning, and reminding.

The key features of the violation target extracted in Section 5 above are used as the inference condition domain. By using the method of semantic association depth confidence network, the violation and normal condition detection are implemented separately. The difference judgment criterion between violation and the normal condition is given by combining the actual requirements or expert experience, which can obtain the judgment result as the inference decision domain. The rule inference model is established as follows.

Given an input conditional domain of $X$ and an output decision domain of $Y$, for $x \in X$ and $y \in Y$, the basic model of inference is as follows.

Rule $R_1$: If the input $x$ satisfies the condition $A$,

Conclusion: the output $y$ is a decision $B$.

Rule $R_2$: if the input $x$ satisfies $A'$,

Conclusion: the output $y$ is a decision $B' = A' \circ (A \rightarrow B)$.

That is, the conclusion $B'$ can be obtained by synthesizing $A'$ with the inference relation from $A$ to $B$. Define the inference relation matrix $R$ from $A$ to $B$ as:

$$R(x,y) = \mu_{A \rightarrow B}(x,y) = \int_{X \times Y} \mu_A(x) \oplus \mu_B(y)/(x,y) \tag{24}$$

where $A \rightarrow B$ represents the inference relation from $A$ to $B$. Then, the conclusion $B'$ can be obtained by synthesizing $A'$ with the inference relation from $A$ to $B$.

According to the values of the elements for the inference relation matrix $R$, the conclusion $B'$ can be determined.

The information derived from the violation data is used to reason with the model, synthesize the operation process, locate the violation category, and realize the automatic reasoning of the violation status, which can realize the automation of violation detection and danger trend prediction for construction sites. If violations and dangerous trends are

found, it can also automatically generate a sequence of preventive measures, i.e., control orders. The function of reminding management, construction workers, and maintenance personnel pay attention to the status of safe operation is implemented.

The specific inference rules are as follows

According to the matching values of the five index values of gray mean, variance, shape, perimeter, and area of the feature, five thresholds are set and discriminated according to the inference rules. The principle of discrimination is defined here as the following inference rules:

Rule $R_1$: If the five matching values all satisfy the threshold, i.e., the grayscale mean, variance, shape, perimeter, and area meet the threshold conditions, the feature to be examined is a standard class of features and is normal;

Rule $R_2$: If the three features of shape, perimeter, and area are combined, or the four matching values formed by any combination for the two features of mean and variance satisfy the threshold, that is, three of the shape, perimeter, and area, or mean and shape, or if there are four perimeters, area, variance, shape, and perimeter matching values all meet the threshold limit, the features to be checked are standard, with a high probability of being normal. In this case, a reminder needs to be given, and further inspection of other feature identifiers is required;

Rule $R_3$: If any one of the matching values of the three features of shape, perimeter, and area does not satisfy the threshold limit, for example, if the four indicators of gray mean, variance, shape, and perimeter all meet the threshold conditions, the feature to be checked is a certain type of feature, but it is not normal. An alarm alert needs to be given in this case to pay attention to safety.

## 7.2. Experiment and Analysis

For the target images finished by reduction and classification in Sections 3 and 4, the inference rule method given in this section is used to synthesize. Features such as target attributes, adjacent regions, and region boundary shapes are related. The inference relationship matrix $R$ between the two feature sets is calculated according to Formula (22), and finally, the conclusion can be judged. The reconstructed image after the classification is shown in Figure 10.

(a) target area

(b) irrelevant area

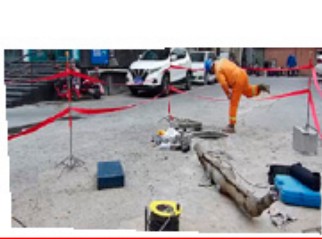
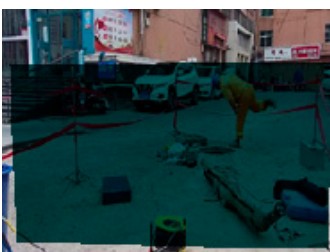

(c) image synthesized by inference

(d) original image

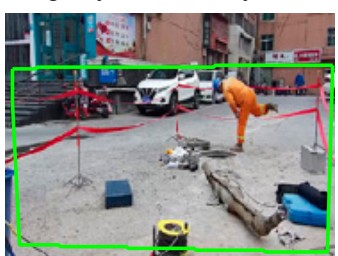
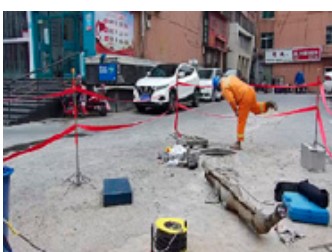

**Figure 10.** Synthesis of images by the inference rule method proposed.

Figure 10a refers to the target region, i.e., the site construction region. Figure 10b indicates the region unrelated to the construction site. Figure 10c indicates the synthetic image of Figure 10a,b, i.e., the image of the target region combined with other regions. Figure 10d is the original image. The red box shows the selected area, the green box shows the area with a large degree of association under the construction site after the synthesis of the image. From Figure 10, it is clear that the inference rule method proposed in this paper reorganizes the implemented classification results, as the reorganized image has the same information as the original image. It shows that the inference rule method given in this paper is effective and feasible.

## 8. Comparison between the Proposed Intelligent Safety Detection System Method and the Existing Safety Detection Method on a Construction Site

The Pytorch framework was chosen for the experiments, the system environment was Windows 11, and the GPU acceleration was CUDA 11.5.1, CUDNN8.3.1_CUDA11.5. The Intel(R) Core (TM) i7-10750H CPU @ 2.60 GHz was used, and the graphics card used was an NVIDIA GeForce RTX 2060 6G for computing. The program script compilation environment was PyCharm, and the program language used was Python. To verify the performance of the safety detection and identification system established in this paper, its application on an actual construction site is as follows:

(1)   **Data source**

The data used for the experiments in this paper come from an actual construction site video stream, which was captured by a mobile company monitoring the screen. The video stream data in three types of environments, namely simple, bright light, and complex, were sampled separately as training and testing sample databases. The following steps are established:

(i)   A total of 200 construction site drawings in different environments from the three types of databases are randomly selected;

(ii)   One of the two drawings of each construction site is randomly selected, totaling 100 drawings, to form the database of experimental training construction maps; the remaining 100 construction maps form the test sample databases;

(iii)   The intelligent detection system method of this paper is used to train 100 drawings in the established construction drawing databases. In total, 100 templates are obtained as the standard set of construction drawings, and each standard set contains 100 marker drawings. The trained marker drawings are deposited into the collected library as training samples.

(2)   **Calculation of correct detection rate**

Further, to verify the superiority of the security detection method proposed in this paper, a comparison between it and existing security detection methods in actual field image detection is given here.

Assuming that a total of $N$ actual field image detection experiments are conducted; the number of correctly detected targets and violations is counted as $n$ through actual field detection. The correct detection rate can be defined as

$$R_c = \frac{n}{N} \tag{25}$$

The basic steps of detection are as follows:

P1. Testing the graph in the test sample databases with the intelligent system detection method in this paper, and matching the obtained test result graphs with the samples in the training graph databases;

P2. Matching the test diagram to be tested with all the diagrams in the training sample databases. According to 100 random selections, the known image that corresponds most closely to the test image, which matches as the detection result and matching tests;

P3. Each graph in the test sample databases is tested 100 times according to steps P1~P2, and the number of correct and incorrect detections are recorded. The correct detection rate is calculated using Formula (25).

(3)    **Result Analysis**

For irrelevant backgrounds or other objects in the scene, the redundant information reduction method proposed in this paper is used to exclude them, which is not in the detection range. For the two construction personnel targets in the scene, the work area of concern is framed with red lines using the target classification method given in this paper. Taking Figure 11 as an example, for the welding work area, the proposed method is used to detect whether the welding personnel is wearing glasses. The targets area below the eyes of the welding person are also detected to see whether sparks are appearing there. According to the matching algorithm given in this paper, matching is carried out between the features with glasses and without glasses. At the same time, matching is carried out between the features with welding sparks and without sparks. According to the inference rules given in this paper, it is determined whether this member of staff has a violation operation or not, and decided whether to give a warning or a reminder. The detection simulations for fence crossing, safety helmet, workwear, and safety belts are shown in Figures 12–15.

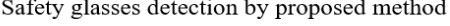

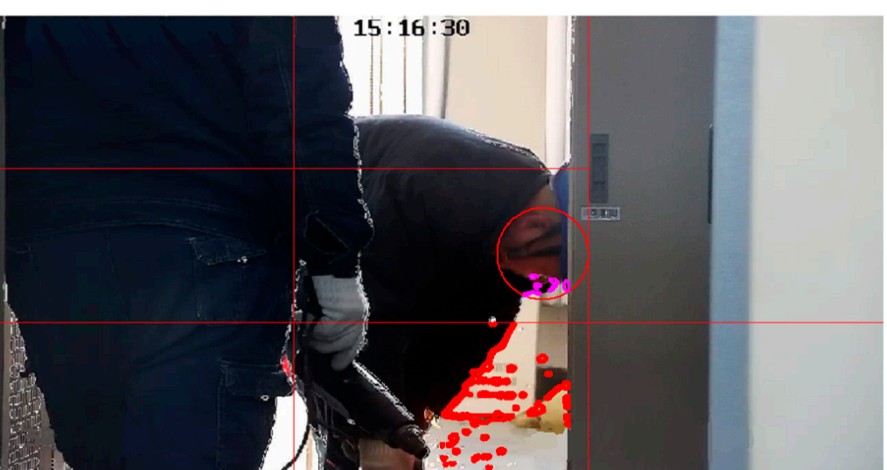

**Figure 11.** Safety glasses detection on the construction site.

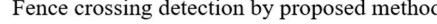

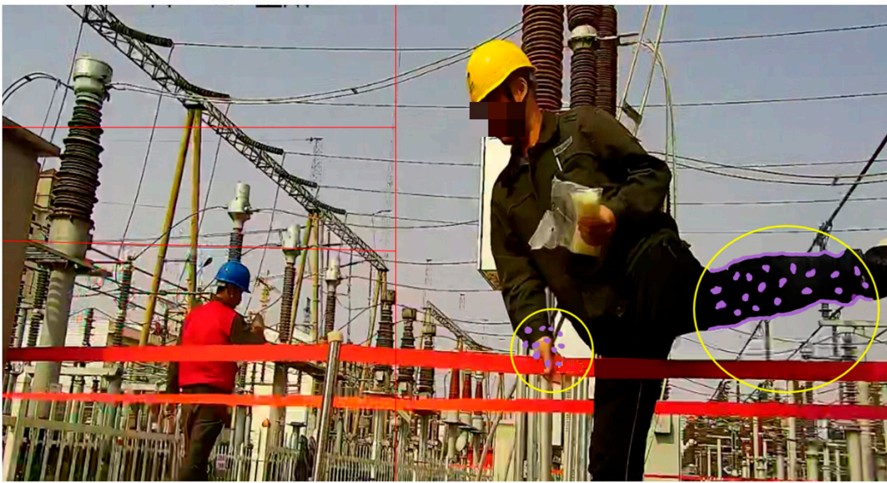

**Figure 12.** Fence crossing detection.

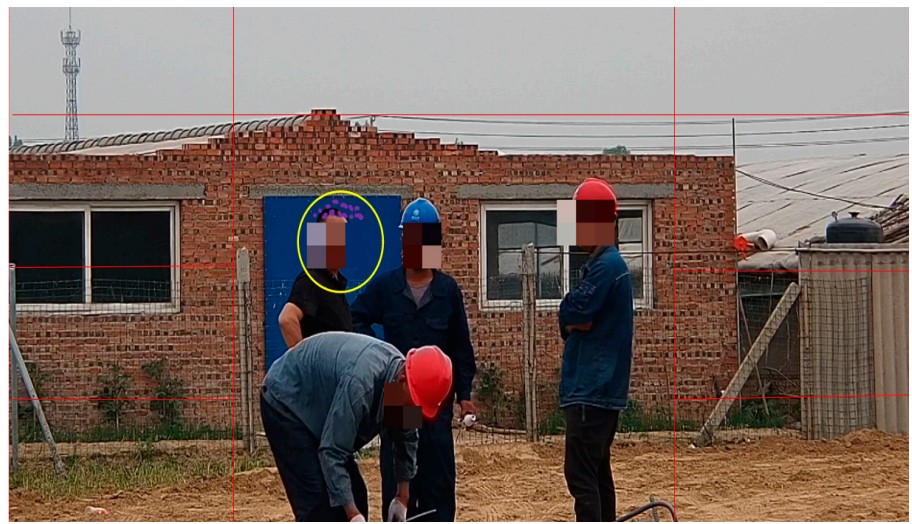

**Figure 13.** Safety helmet detection.

Workwear detection by proposed method

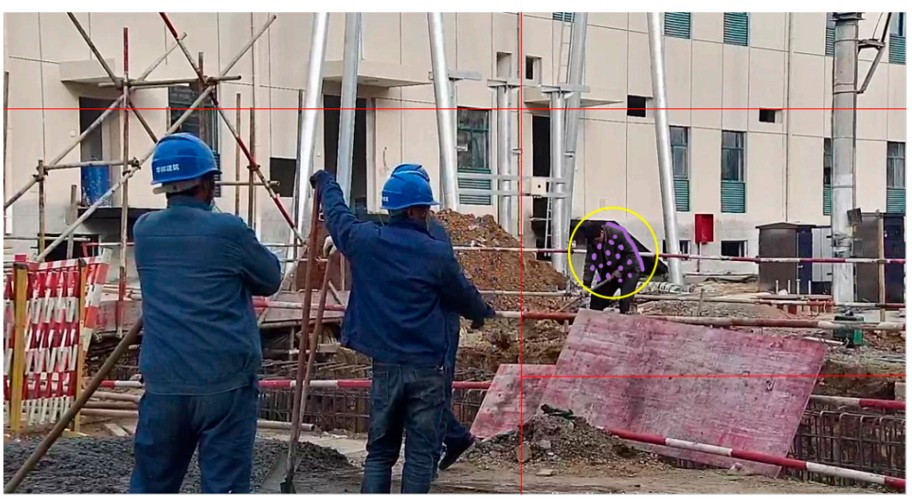

**Figure 14.** Workwear detection.

Safety belts detection by proposed method

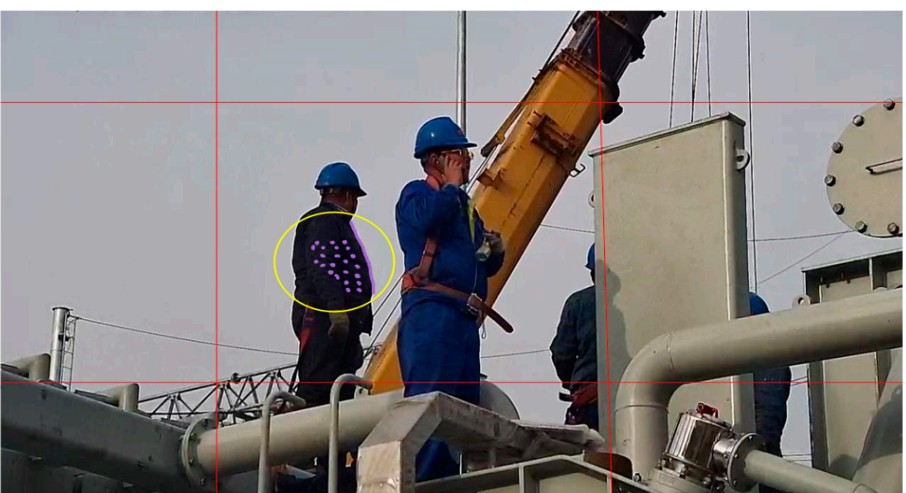

**Figure 15.** Safety belt detection.

From the following five figures, it can be seen that for the empty scenes in the images, the redundant reduction method proposed by this paper does isolate the useless region. In addition, for target classification, the feature extraction, matching, and inference rule methods are used. From the inference rules given in this paper, it is judged that the worker is operating in violation of the law and the program has given him a warning reminder to arrive at the command. The circle in the figure indicates that a violation has been detected in the area, with the focus marked on detection. There are other examples, but some privacy issues are involved, and the function of safety detection has been demonstrated in the following five figures. Therefore, more examples are not necessary. The good results demonstrated from this practical application show that the security check system proposed in this paper is effective and feasible.

In the experiments, 50 samples were experimented with to implement the detection of VoRR. The proposed and existing detection methods [1,5,8] are compared under the repeated execution of 100 experimental cases. For the original image in Figure 11, the correct detection rate of the proposed security detection method is 91.20% on average, the correct detection rate of the YOLOv5 detector [5] is 88.23% on average, the average of deep transfer learning [1] is 79.25%, and the average of feature fusion [8] is 71.63%. As the number of training repetitions increases when each detection method is iterated, feedback is updated and optimized, resulting in improved accuracy in detecting different methods. The change in the correct detection rate with increasing repetitions is shown in Figure 16 and Table 2, which demonstrate the comparison of the correct detection rate of the proposed and existing security detection methods. In addition, the detection speed of the algorithm in this paper is 232.66 ms. However, that of the YOLOv5 detector algorithm is 295.34 ms, that of the deep transfer learning algorithm is 412.98 ms, and that of the feature fusion algorithm is 449.27 ms.

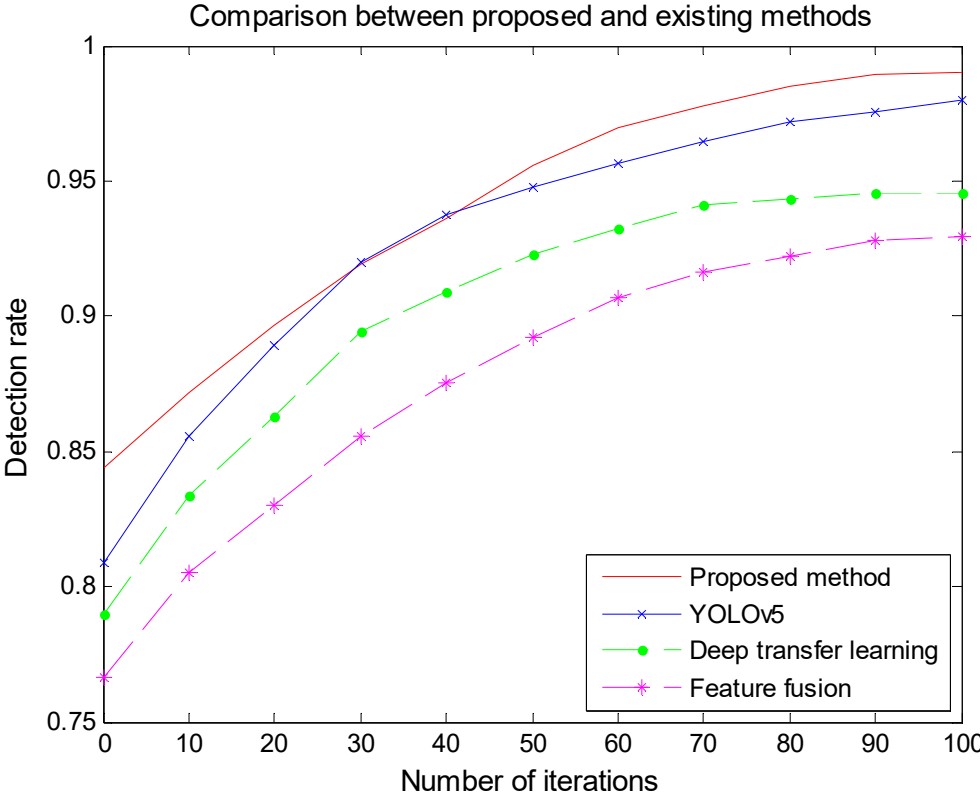

**Figure 16.** Comparison of the correct detection rate of the proposed and existing security detection methods.

**Table 2.** Comprehensive comparison of proposed and existing security detection methods.

| Several Detection Methods | Average Correct Rate (%) | Detection Speed (ms) | Noise Resistance | Adaptation to Environment |
| --- | --- | --- | --- | --- |
| Proposed method | 91.20 | 232.66 | Strongest | Most complex |
| YOLOv5 | 88.23 | 295.34 | Stronger | More complex |
| Deep transfer learning | 79.25 | 412.98 | Stronger | General |
| Feature fusion | 71.63 | 449.27 | Weak | General |

As shown in Figure 16 and Table 2, the safety detection method proposed in this paper is better than the existing methods in terms of the correct detection rate of VoRR. The detection speed is fast, the noise resistance is strong, and the robustness is best. It adapts to the security detection of complex situations in construction scenes, which shows that the detection method proposed in this paper has good comprehensive performance. The time complexity is dependent on the complexity of Formula (25) and the three methods compared in this paper depend on the network structure. The time complexity is close to 1 s and the proposed method is less than 0.5 S. Because the method proposed in this paper only involves the algorithm itself, the complexity index degree is relatively small. The correct rate and detection speed here are calculated from the actual detection of each method after the target program is run. However, the noise resistance and adaptability to the environment are estimated according to the actual situation so that each method can correctly detect the target during the experiment.

## 9. Conclusions

To solve the problem of automatically completing the detection of high-speed image streams and giving reminders for construction safety on site, this paper proposes a violation of rules and regulations recognition method on construction sites, and gives a matching method by automatically obtaining a few samples. Based on the analysis and classification of high-speed image streams, cameras with different positions and angles are set up to implement high-resolution image stream acquisition. A correlation degree method of image redundant information reduction is proposed. After the redundant information reduction, the image background and some image target components unrelated to construction are filtered out, while the construction personnel information and construction personnel operation tools are retained. The nearest neighbor classification algorithm is established, and almost all the regions with the features of the targets of interest are classified, which can allow violations to be identified more precisely. Large targets at construction sites are classified according to the requirements of target features or rules, the selected targets are extracted with little information loss, and well-detailed features are maintained. The feature extraction method of singular value decomposition is given, which extracts the features of the target information from the image and calculates the feature vector of the target to be detected. The matching method of the correlation matrix is proposed, which is to perform difference matching or similarity, which matches between the features of the standard library and the target features to be examined. Using the output value, the feature calculation layer correlates and classifies diverse features, different indicators, and true/false features, which can match the target parameter vector. The inference rule model is established, which is the criterion for identifying the target and judging the violation. Building the output value of matching layer, implement security detection, warning and alert functions for violation determination. The experimental results show that the accuracy rate of the safety detection method proposed in this paper is as high as 90%. Compared with existing safety detection methods, it has the best comprehensive performance for security warning and alarm functions. This research will provide a new method of decision support, target detection, and recognition in multiple different scenarios.

This paper only focuses on the wearing of helmets and the detection of irregularities during welding, which has certain limitations. Firstly, the method in this paper needs to be

further enhanced for cases of distortion or occlusion caused by camera acquisition. The method cannot accurately identify the possible hazards of objects moving at high speed. In addition, construction site scenes are complex and violations are diverse, so subsequent research will classify construction scenes. The behavior in each scene will be detected safely for the application of the simultaneous monitoring of multiple scenes. This method is more severe and complex, so further in-depth research will be carried out in future papers to promote universality and wide application.

**Author Contributions:** Q.W. and W.W. wrote the main manuscript text and prepared Figures 11–16. H.C. and L.Z. prepared Figures 1–5. Y.L. and X.Q. prepared Figures 6–10. All authors have read and agreed to the published version of the manuscript.

**Funding:** This work is supported by the Key Science and Technology Program of Henan Province (222102210084), Key Science and Technology Project of Henan Province University (23A413007), and National Natural Science Foundation of China Project (62076223), respectively.

**Institutional Review Board Statement:** All authors agree that the research in this paper does not involve theoretical studies of humans and animals. The figures in the paper were provided by QingE Wu and consent has been obtained from the people in the figures. All authors agree to make this paper publicly accessible.

**Informed Consent Statement:** Written informed consent has been obtained from the participants to publish this paper.

**Data Availability Statement:** All data generated or analyzed during this study are included in this published article. We do wish to share our raw data, and these data are original.

**Conflicts of Interest:** This manuscript has not been published and is not under consideration for publication elsewhere. We have no conflict of interest to disclose.

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
