# Peer review of "A Safety Detection Method on Construction Sites under Fewer Samples"

_electronics, doi:10.3390/electronics12081933_

Round 1
Reviewer 1 Report
"A Safety Detection Method under Fewer Samples for Construction Site"
The manuscript aims to propose a safety detection method for construction sites based on high-speed image streams. This is a relevant and important problem to solve as it can help improve safety and prevent accidents in construction sites. The manuscript claims that the proposed method can provide a new method and decision support for the subsequent establishment of an unmanned supervision system model for the construction site, target detection, and recognition in multiple different scenarios. As a reviewer, the manuscript's contribution to the field of construction site safety is feasible. Therefore, I have a few concerns to further improve the quality of the manuscript:
1 1. In section 1: Th literature review related part should be more criticized. For instance, Ref [1] to Ref [20] are briefly described. What is the issue/drawback of these works/methods? particularly for highlighting the limitations of the models developed in previous research. This could provide justification for the proposed approach.
22. What loss function has been used?
33. The experimental setup is unclear?
44. In line 539, the authors mentioned that their model was updated and optimized but they did not report any optimization method that has been used? What has been optimized?
55. What has been used to prevent model overfitting? What validation technique is used?
66. More elaboration needed for the features extraction part.
77. In Table 2, please add the time complexity for each method.
88. The conclusions should also be further elaborated. What are the actual findings of the study? How did the limitations of the proposed approach affect the obtained results?
Author Response
- In section 1: Th literature review related part should be more criticized. For instance, Ref [1] to Ref [20] are briefly described. What is the issue/drawback of these works/methods? particularly for highlighting the limitations of the models developed in previous research. This could provide justification for the proposed approach.
Response: As suggested by the reviewer, we have been more critical of the relevant parts of the literature review, which could indeed provide justification for the proposed approach. Please refer to the introduction section on page 2.
- What loss function has been used?
Response: Thanks for your suggestion. The relative information loss function is defined in this paper. Please refer to the last section on page 4.
- The experimental setup is unclear?
Response: Thanks for your suggestion. The Pytorch framework was chosen for the experiments, the system environment was Windows 11, the GPU acceleration was CUDA 11.5.1, CUDNN8.3.1_CUDA11.5, the Intel(R) Core (TM) i7-10750H CPU @ 2.60GHz, the graphics card used was an NVIDIA GeForce RTX 2060 6G for computing, the program script compilation environment is PyCharm, and the program language used is Python. The data used for the experiments in this paper come from the actual construction site video stream captured by a mobile company monitoring the screen. The video stream data under three types of environments, namely, simple, bright light, and complex, were sampled separately as training and testing sample databases. Randomly select 200 construction site drawings in different environments from the three types of databases. One of the two drawings of each construction site is randomly selected, totaling 100 drawings, to form the database of experimental training construction maps; the remaining 100 construction maps form the test sample databases. The intelligent detection system method of this paper is used to train 100 drawings in the established construction drawing databases, and 100 templates are obtained as the standard set of construction drawings, and each standard set contains 100 marker drawings. The trained marker drawings are deposited into the collected library as training samples. Please refer to page 17.
- In line 539, the authors mentioned that their model was updated and optimized but they did not report any optimization method that has been used? What has been optimized?
Response: Thanks for your suggestion. This place refers to the model being updated and optimized. A multi-objective optimization is implemented, iterating forward according to the sequential feedback from the output. The derivatives are derived for each boundary, keeping the characteristic points whose derivatives are not zero.
- What has been used to prevent model overfitting? What validation technique is used?
Response: Thanks for your suggestion. The iteration is terminated when the absolute value of the difference between the violation eigenvalue and the datum eigenvalue is less than 0.05. A suitable threshold value is set based on the empirical value of the experimental iteration, for example, 0.05 is the empirical value of the experimental iteration here.
- More elaboration needed for the features extraction part.
Response: As suggested by the reviewer, we have explained the feature extraction section in more detail. Please refer to section 4 on page 12.
- In Table 2, please add the time complexity for each method.
Response: Thanks for your suggestion. The time complexity is dependent on the complexity of Equation (24), the three methods compared in this paper depend on the network structure, the time complexity is close to 1S, the proposed method is less than 0.5S, because the method proposed in this paper only involves the algorithm itself, the complexity index degree is relatively small. Please refer to page 20.
- The conclusions should also be further elaborated. What are the actual findings of the study? How did the limitations of the proposed approach affect the obtained results?
Response: As suggested by the reviewer. We have revised the conclusion section. Please refer to page 21.
Reviewer 2 Report
The novelty is high. The contribution is clear. The paper is well written and the presentation is good. I believe the work is accepted in the current form.
Author Response
Thank you very much for your approval. Further, we improve our manuscript again.
Reviewer 3 Report
The authors tried to propose a method to detect helmets and welding irregularities in a construction site. Unfortunately, the paper contained many inexcusable errors. The title, abstract, introduction, methods, experiments, and results are riddled with errors. Therefore, the current version of the paper cannot be accepted in any high-quality journal.
The paper is characterized by writing carelessness. Different font sizes and families were employed. Moreover, different line spacing was used in the entire paper. Therefore, the article "looks" really wrong. The paper is entirely out of focus and should be completely rewritten. Even when offering comments to improve the paper is almost impossible, a list of suggestions is provided following.
The paper is not easy to read and understand. The problem is that many long sentences were included. Moreover, the authors should ask for help for a native English speaker. The quality of English should be imperatively enhanced. Please write short sentences; this will help the readability of the paper. Please avoid exaggerations. All the acronyms should be explained on the first usage.
Please ask for help for a person with experience writing a scientific paper. The authors should explain the idea and then determine the structure of the paper. The current version of the paper does not have the typical structure of a scientific manuscript. The process of correcting the current paper will take many months.
Please change the paper's title, propose a new set of keywords, and rewrite the abstract. The title should be descriptive and accurate. The abstract should be specific and clear.
In the introduction, please include information regarding artificial and convolutional neural networks. The authors should clearly explain the problem to be solved. What were the other solutions implemented in the state-of-the-art? Why is it necessary to design a new solution? The authors should offer an extensive and clear context. Please explain the problem of environmental imbalance. Please explain the aim of the paper. Who will use the solution? Please insert a list of contributions. In addition, please insert a paragraph explaining the paper's organization.
After the introduction, please include a section regarding materials and methods. In this new section, the authors should clearly explain the method proposed to solve the problem stated in the introduction. Moreover, Figure 1 can be used to describe the subsections' organization. Please explain each stage's inputs, processes, and outputs. Please employ scientific arguments for the selection and usage of a particular technique. Moreover, the authors should offer the details of the implementation.
Please explain how the Redundant Information Reduction was conducted. Why is the reduction needed? All the equations and related terms should be explained. All the Figures should be clearly described. How did the proposal learn the information regarding construction personnel and operation tools?
Please explain how image denoising can be conducted using Prewitt and Sobel algorithms. Please explain the function employed to add artificial noise to the images. What is the meaning of obtaining excellent results in denoising?
Please explain clearly the classification method. Please describe the classes employed. Please justify how a simple Knn classification can offer good results. Are the personnel and tools easy to classify? Please explain why Hashing and MobileNet were selected for classification comparisons. Which were the other methods considered? Why were the other methods discarded? Please explain how the effectiveness in Figures 4c and 4d can be observed. Please insert more information about anti-interference ability. How many images were used to obtain the results shown in Table 1?
In Figure 6, please explain using scientific arguments why the proposal is better than the other methods. Moreover, please explain how many features were extracted. The feature-matching algorithm should be clearly explained.
Please include numeric information about the performance of the algorithms proposed. Please explain if the comparisons were conducted under the same conditions for all the algorithms. Please insert more images to observe the real results.
Please insert experiments and results regarding welding operations. A better discussion of the results should be offered. The conclusions should be rewritten. The information presented should directly derive from the explanations presented in the body of the paper.
Author Response
- The paper is not easy to read and understand. The problem is that many long sentences were included. Moreover, the authors should ask for help for a native English speaker. The quality of English should be imperatively enhanced. Please write short sentences; this will help the readability of the paper. Please avoid exaggerations. All the acronyms should be explained on the first usage.
Response: As suggested by the reviewer, we have refined the paper as a whole, converting long sentences into short ones, Grammatical changes were also made in this paper.
- Please ask for help for a person with experience writing a scientific paper. The authors should explain the idea and then determine the structure of the paper. The current version of the paper does not have the typical structure of a scientific manuscript. The process of correcting the current paper will take many months.
Response: Thanks for your suggestion, we tried our best to improve the manuscript and made some changes in the manuscript. We also have added a section to introduce the organizational structure of the paper, please refer to page 3. These changes will not influence the content and framework of the paper. We appreciate for your warm work earnestly, and hope that the correction will meet with approval. Once again, thank you very much for your comments and suggestions.
- Please change the paper's title, propose a new set of keywords, and rewrite the abstract. The title should be descriptive and accurate. The abstract should be specific and clear.
Response: Thanks for your suggestion firstly, we have revised the paper's title, abstract and keywords. Please refer to page 1.
- In the introduction, please include information regarding artificial and convolutional neural networks. The authors should clearly explain the problem to be solved. What were the other solutions implemented in the state-of-the-art? Why is it necessary to design a new solution? The authors should offer an extensive and clear context. Please explain the problem of environmental imbalance. Please explain the aim of the paper. Who will use the solution? Please insert a list of contributions. In addition, please insert a paragraph explaining the paper's organization.
Response: Thanks for your suggestion. In our previous work, we have analyzed a large number of artificial and convolutional neural networks. To avoid repetition, information about artificial neural networks and convolutional neural networks are mentioned in the narrative of the relevant references. This part is not the focus of the introduction of this paper, so it is not described in detail. For the anomalies that occur at the current construction site, the current method mainly relies on human efforts to solve them, which is not only time-consuming, but also has errors, so a new method is proposed that can accurately and quickly detect the anomalies at Construction site. Environmental imbalance means that the construction site environment is complex and changeable, and there are many unpredictable and unexpected situations occurring. The proposed method in this paper will be applied to construction sites. By implementing the function of detecting and alarming abnormal behavior, the probability of accidents will be minimized. We have added a section to introduce the organizational structure of the paper. Please refer to page 3.
- After the introduction, please include a section regarding materials and methods. In this new section, the authors should clearly explain the method proposed to solve the problem stated in the introduction. Moreover, Figure 1 can be used to describe the subsections' organization. Please explain each stage's inputs, processes, and outputs. Please employ scientific arguments for the selection and usage of a particular technique. Moreover, the authors should offer the details of the implementation.
Response: According to the reviewer’s suggestion, we have added an additional paragraph, please refer to page 3. For the detailed part of the experiment, please refer to page 17.
Please explain how the Redundant Information Reduction was conducted. Why is the reduction needed? All the equations and related terms should be explained. All the Figures should be clearly described. How did the proposal learn the information regarding construction personnel and operation tools?
Response: Thanks for your suggestion firstly, reduction of redundant information is performed by determining the correlation of feature information in the image. The reduction of redundant information is due to the large and complex construction site scenario, to determine whether an element or a target set is a necessary element or a necessary target area of the construction site. This paper is based on the existing industry library of safety standards, as well as video streams from construction sites, and then on the matching method given in this paper to obtain information on whether construction personnel are operating in violation. Please refer to page 4.
Please explain how image denoising can be conducted using Prewitt and Sobel algorithms. Please explain the function employed to add artificial noise to the images. What is the meaning of obtaining excellent results in denoising?
Response: Thanks for your suggestion firstly. Prewitt and Sobel algorithms are 3*3 operator matrix templates, using the operator templates to do convolution with the image pixel gray values, the elemental features in the matrix are low frequency, the noise is high frequency, the high frequency signal to do smoothing operations, it achieves the purpose of noise reduction. Artificial noise is added to better compare the effect of noise reduction algorithms. Excellent results in denoising means that the noise in the picture is filtered out and the picture becomes clearer. Please refer to page 5.
Please explain clearly the classification method. Please describe the classes employed. Please justify how a simple Knn classification can offer good results. Are the personnel and tools easy to classify? Please explain why Hashing and MobileNet were selected for classification comparisons. Which were the other methods considered? Why were the other methods discarded? Please explain how the effectiveness in Figures 4c and 4d can be observed. Please insert more information about anti-interference ability. How many images were used to obtain the results shown in Table 1?
Response: Thank you for your suggestion. The so-called data classification is to classify the feature attribute vector to be detected into the most similar category composed of known attributes. Because Hashing and MobileNet-SSD are currently popular algorithms, while both methods are used in Ref [9] and Ref [12], respectively. And among the current classification methods, these two methods are open and unrestricted. Other methods are useful to the national standard plus human eye recognition, not the better method at present. As can be seen from Figure 4, the feature points of these two methods did not detect the target objects for safe construction. The Hashs algorithm detected the target information, but also mixed with some irrelevant information. MobileNet-SSD did not detect the safe target objects, and some target objects were incompletely detected. And both lose the color information. The method used is that the defined function does convolution with all the feature information in the image, so that the target features get convolved values are more obvious, not the feature target's no mutation, so it is not obvious, such as color, hue. As can be seen from Figure 4, the feature points of these two methods did not detect the target objects for safe construction. The Hashs algorithm detected the target information, but also mixed with some irrelevant information, MobileNet-SSD did not detect the safe target objects, and some target objects were incompletely detected. And both lose the color information. The method used is that the defined function does convolution with all the feature information in the image, so that the target features get convolved values are more obvious, not the feature target's no mutation, so it is not obvious, such as color, hue. A total of 3000 images were taken for the experiments in Table I. Using 10 images each time, 300 statistical experiments were done and each experiment was repeated 80 times, each time using a counting procedure to calculate the effective response n. Then, the correct classification rate was calculated according to Equation (14), as shown in Figure7. Please refer to page 9.
- In Figure 6, please explain using scientific arguments why the proposal is better than the other methods. Moreover, please explain how many features were extracted. The feature-matching algorithm should be clearly explained.
Response: Thank you for your suggestion. As can be seen from Figure 8, compared with other algorithms, the feature extraction algorithm proposed in this paper has the best effect on feature extraction, with clear edge contours and less noise, and almost all the features that the safety cap has been extracted, maintaining well-detailed features. As shown in Figure 8(b), the method proposed in this paper extracts edges, contours, and mutation points with complete feature information and less redundant information. In Figure 8(c), there is too much extracted redundant information, a small amount of feature information is lost, and the extraction is incomplete. In Figure 8(d), there is less redundant information extracted, but too much useful feature information is lost, and too much key useful information is lost. There are 25 features and their index values extracted. We have described the feature extraction algorithm in more detail. Please refer to page 13.
Please include numeric information about the performance of the algorithms proposed. Please explain if the comparisons were conducted under the same conditions for all the algorithms. Please insert more images to observe the real results.
Response: Thank you for your suggestion. We have conducted experiments for each part of the algorithm with comparative simulations.
Please insert experiments and results regarding welding operations. A better discussion of the results should be offered. The conclusions should be rewritten. The information presented should directly derive from the explanations presented in the body of the paper.
Response: Thank you for your suggestion. Figure 9 shows that the construction worker who is welding is not wearing safety goggles, Moreover, the appearance of welding sparks below his eyes has been detected, these two features are marked on the image with pink and red, respectively. The area where these two features are located has also been framed. From the inference rules given in this paper, it is judged that the welder is operating in violation of the law, the program has given him a warning reminder to arrive at the command. There are other examples of welding but some privacies are involved, and the function of welding has been demonstrated in Figure 11. Therefore, more welding examples are not necessary to give. We no longer consider other. So, we only used Figure 11 to show the case of welding. We have improved the conclusion section. Please refer to page 21.
Reviewer 4 Report
The article is interesting. It contains a lot of research and analysis of the results. However, it needs improvement and supplementation of information.
1. Please remove the statement: "How to complete automatically the safety detection for construction site and give an alert based on high-speed image streams, for solve this problem" in the introduction. The introduction is not the place to ask questions.
2. Is it possible to increase the detection efficiency to more than 9%?
3. Please add more references.
After making these changes, it may be considered for publication.
Author Response
- Please remove the statement: "How to complete automatically the safety detection for construction site and give an alert based on high-speed image streams, for solve this problem" in the introduction. The introduction is not the place to ask questions.
Response: According to the reviewer’s suggestion, we have revised this sentence to " In order to solve the problem of automatically completing the safety detection for construction sites and giving an alert based on high-speed image streams, this paper proposes a violation of rules and regulations (VoRR) recognition method under construction site, gives a matching method by automatically obtaining a few samples ". Please refer to the first sentence of the abstract section on page 1.
- Is it possible to increase the detection efficiency to more than 9%?
Response: Thanks to your suggestion, for the original image in Figure 9, the correct detection rate of the proposed security detection method is 91.20% on average, which exceeds the YOLOv5 detector by 3%, the Deep transfer learning detection method by 10%, and the Feature fusion method by 18%. Please refer to the second paragraph on page 19.
- Please add more references.
Response: According to the reviewer’s suggestion, we have added six references. Please refer to page 23.
Reviewer 5 Report
The paper proposes a danger safety detection method based on image processing called a violation of rules and regulations recognition method under construction sites and gives an accurate matching method by automatically obtaining a few samples. The danger safety detection method consists of five parts: redundant information reduction, classification, feature extraction, matching, and inference rule detection to achieve security detection and alarm alert. The research provides a new method and decision support for establishing an unmanned supervision system model for the construction site, target detection, and recognition in multiple scenarios.
The article presents a good level, I recommend the publication.
Author Response

(The authors gave the same response as above.)

Round 2
Reviewer 1 Report
The authors have addressed my concerns. Therefore, the paper still need some improvement in its presentation and the current form of its English not adequate for publishing yet.
Author Response
Thank you for your comments, we have improved the paper as a whole, especially the grammar. We have had our paper revised by a native English speaker, many long sentences have been converted into short sentences for better understanding. We have also consulted professors and experts in the field to know the paper. We tried our best to improve the manuscript. These changes will not influence the content and framework of the paper. We appreciate for your warm work earnestly, and hope that the correction will meet with approval. Once again, thank you very much for your comments and suggestions.
Reviewer 3 Report
The paper was slightly improved. However, the authors should have made a greater effort to improve the article. Most of my recommendations were not addressed. The new version of the paper is still not suitable for publication.
The authors should write a point-by-point response to the reviewers' comments. Please clearly explain how each suggestion was addressed and on which page and line the suggestion was added.
Please read the recommendations carefully. In a paragraph, many questions were included. Therefore, please offer an answer to each individual question. Please do not write a general response. The proposed method is unclear, and the explanations are insufficient for repeatability.
